# In Silico Evaluation of Sesquiterpenes and Benzoxazinoids Phytotoxins against M^pro^, RNA Replicase and Spike Protein of SARS-CoV-2 by Molecular Dynamics. Inspired by Nature

**DOI:** 10.3390/toxins14090599

**Published:** 2022-08-29

**Authors:** Francisco J. R. Mejías, Alexandra G. Durán, Nuria Chinchilla, Rosa M. Varela, José A. Álvarez, José M. G. Molinillo, Francisco García-Cozar, Francisco A. Macías

**Affiliations:** 1Allelopathy Group, Department of Organic Chemistry, Institute of Biomolecules (INBIO), Campus CEIA3, School of Science, University of Cádiz, C/República Saharaui, 7, 11510 Puerto Real, Spain; 2Center for Molecular Biosciences (CMBI), Institute of Pharmacy/Pharmacognosy, University of Innsbruck, 6020 Innsbruck, Austria; 3Department of Physical Chemistry, Faculty of Sciences, INBIO, University of Cádiz, 11510 Puerto Real, Spain; 4Department of Biomedicine, Biotechnology and Public Health, University of Cádiz and Institute of Biomedical Research Cádiz (INIBICA), 11009 Cádiz, Spain

**Keywords:** molecular dynamics, docking, SARS-CoV-2, COVID-19, sesquiterpene, benzoxazinoid

## Abstract

In the work described here, a number of sesquiterpenes and benzoxazinoids from natural sources, along with their easily accessible derivatives, were evaluated against the main protease, RNA replicase and spike glycoprotein of SARS-CoV-2 by molecular docking. These natural products and their derivatives have previously shown remarkable antiviral activities. The most relevant compounds were the 4-fluoro derivatives of santamarine, reynosin and 2-amino-3*H*-phenoxazin-3-one in terms of the docking score. Those compounds fulfill the Lipinski’s rule, so they were selected for the analysis by molecular dynamics, and the kinetic stabilities of the complexes were assessed. The addition of the 4-fluorobenzoate fragment to the natural products enhances their potential against all of the proteins tested, and the complex stability after 50 ns validates the inhibition calculated. The derivatives prepared from reynosin and 2-amino-3*H*-phenoxazin-3-one are able to generate more hydrogen bonds with the M^pro^, thus enhancing the stability of the protein–ligand and generating a long-term complex for inhibition. The 4-fluoro derivate of santamarine and reynosin shows to be really active against the spike protein, with the RMSD site fluctuation lower than 1.5 Å. Stabilization is mainly achieved by the hydrogen-bond interactions, and the stabilization is improved by the 4-fluorobenzoate fragment being added. Those compounds tested in silico reach as candidates from natural sources to fight this virus, and the results concluded that the addition of the 4-fluorobenzoate fragment to the natural products enhances their inhibition potential against the main protease, RNA replicase and spike protein of SARS-CoV-2.

## 1. Introduction

Coronaviruses (CoVs) are large positive-strand, enveloped non-segmented RNA viruses that generally cause enteric and respiratory illnesses in animals and humans [1]. Although most CoVs that affect humans produce only mild respiratory diseases, with little or no mortality, the previous epidemics of two pathogenic CoVs, namely severe acute respiratory syndrome coronavirus (SARS-CoV) and Middle East respiratory syndrome coronavirus (MERS-CoV), led to major health alerts.

Traditional medicine based on plants has been used for preventive treatments for COVID-19 in countries all over the world. Furthermore, some nutrient supplements obtained from herbal sources have also proven effective in reducing virus transmission and decreasing infection [2]. Among the families of compounds that are potential drugs in traditional medicine, sesquiterpenes are relevant due to their broad-spectrum drug nature, e.g., artemisinin, and this family of compounds is found in *A. annua*. The in vitro efficacy of artemisinin-based treatments in combating SARS-CoV-2 has shown that treatment with artesunate, artemether, *A. annua* extracts and artemisinin hindered viral infections of human lung cancer A549-hACE2 cells, VeroE6 cells and human hepatoma Huh7.5 cells. Among these four treatments, artesunate showed the strongest anti-SARS-CoV-2 activity (7–12 μg/mL) [3,4]. Given the promising results obtained with terpenoids, in silico evaluation seems to be a promising tool to select leads for future bioassays against SARS-CoV-2.

Previous in silico studies have demonstrated the efficacy of natural products fighting against SARS-CoV-2. A plant-derived alkaloid, such as cryptoquindoline and 6-oxoisoiguesterin isolated from *Cryptolepis sanguinolenta* and *Salacia madagascariensis*, displayed inhibition against the M^pro^ [5]. Forrestall et al. also evaluated the activity against the M^pro^ by molecular docking of different natural products with 2-pyridone scaffolds, mainly based on diterpene skeletons [6]. On the other hand, Narkhede et al. did not use a skeleton criterion and selected different kinds of natural products with previous antiviral activity [7]. In the same case, compared to the other, neither sesquiterpene nor benzoxazinoids have been studied in depth by molecular dynamics and docking.

The work described here concerned the evaluation of inhibitors for the three main targets of SARS-CoV-2 (M^pro^ [8,9], spike glycoprotein [10,11] and RNA replicase [12,13]) by molecular docking and molecular dynamics simulation studies on bioactive natural products and derivatives obtained from natural sources. These compounds can be obtained on a multigram scale or can be synthesized in a single step, and they are readily available and are relatively inexpensive.

## 2. Results and Discussion

### 2.1. Molecular Docking Studies

A total of 12 sesquiterpene lactones and 14 benzoxazinoids (Figure 1A,B) were selected from the natural products and derivatives with notable bioactivity and structural similarity to the reference standards (Figure 1C). The results for these compounds were compared with those obtained for the standards. All of the compounds have previously shown anti-cancer activity (mostly anti-leukemia) or some other cytotoxicity [14,15,16]. Antiviral activity is also displayed, as in the case of costunolide, DHC and alantolactone, against the Hepatitis C virus [17]. Inhibition of this virus has been also observed after the application of artichoke extracts containing cynaropicrin [18]. APO and different benzoxazinoids present activity against human cytomegalovirus and herpes simplex virus type 1 [19,20]. Favipiravir and hydroxychloroquine contain two fused rings with at least one heteroatom in the structure, as do the benzoxazinoids **DIBOA**, **DIMBOA**, **DDIBOA** and **APO**. In addition, the presence of a halogen in the structures of the standards inspired us to include 4-fluorobenzoate derivatives in the study. Methyl 4-fluorobenzoate (**Met-4F-Benzo**) was included in the test in order to ascertain whether the activity can be attributed to this fragment alone. In contrast, artemisinin is an antimalarial compound isolated from Artemisia annua, and this is already being tested [21,22]. Artemisinin has a lactone sesquiterpene skeleton (C-15 and cyclic ester in the main structure), as do the costunolide, dehydrocostuslactone (DHC), cynaropicrin and alantolactone (alanto) derivatives. Azithromycin was included in the study as a negative standard due to its different backbone and its reported lack of efficacy against COVID-19 disease [23].

The binding energies of the sesquiterpenoids toward the M^pro^, RNA replicase and the spike protein of SARS-CoV-2 in comparison with the standards are provided in Table 1. 

The binding energy values show the remarkable activity of artemisinin, which has not been tested previously, on all of the proteins tested. Furthermore, artemisinin has similar binding energies to costunolide and **DHC**, two compounds isolated on a multigram scale from *Saussurea Lappa* (Decne.) Sch.Bip [24]. Nevertheless, the highest activities were obtained for the 4-fluorobenzoate derivatives of reynosin and santamarine (**Fluor-Reynosin** and **Fluor-Santamarine**). In terms of the M^pro^ and RNA replicase inhibition values (Table 1), artemisinin gave values in the range 25–35 μM, while **Fluor-Reynosin** and **Fluor-Santamarine** were in the range 1–20 μM. The results of the studies on the spike protein are consistent with the recognition function that this receptor protein has. In this case (Table 1), the bis (4-fluorobenzoate) derivative of cynaropicrin (**Fluor-Cynaro**) was the most active, with an inhibition constant of 1.10 μM on the spike protein.

Small changes in the skeleton did not result in significant changes in the binding energy. A comparison of the results for reynosin, santamarine, alantolactone (**alanto**), β-cyclocostunolide (**beta-cyclo**), α-cyclocostunolide (**alpha-cyclo**) and 3-deoxybrachylaenolide (**3-DeBra**) clearly shows that the arrangement of the skeleton does not lead to changes in the inhibition in computational studies and even the presence of a hydroxyl group or double bond in the first ring of the structure did not alter the energy markedly. An analysis of the ligand binding site and the intermolecular forces (Appendix A and Table 1) indicated that the lactone group appears to be the main component required for activity. Nevertheless, **alanto** displayed a significant binding value, which was better than those for similar lactones, against the RNA replicase. **Alanto** differs from the other sesquiterpenes in the lactone arrangement, and this indicates that the remaining carbon skeleton must play a relevant role.

As far as the benzoxazinoids (Table 2) are concerned, the results are similar to those described for the sesquiterpenoids. These compounds all showed a binding energy toward the M^pro^ that was lower than that of the standard artemisinin, but they are more active than the standards with similar skeletons (hydroxychloroquine and favipiravir). In addition, the 4-fluorobenzoate derivative of APO (**Fluor-APO**) has values similar to artemisinin. The RNA replicase shows different profiles, with APO and the 4-fluorobenzoate derivative of 2,2′-disulfanediyldianiline (**Fluor-DisNH**) being more active than they were against the M^pro^. The spike protein did not seem to recognize this kind of skeleton easily, but the presence of halogen atoms (Appendix A) linked at the edge of the fluorobenzoate fragment does appear to be relevant.

Previous studies regarding similar proteins have highlighted the efficacy of this kind of compound. Xue et al. showed that Michael acceptors groups in molecules, with the same function as sesquiterpenes lactones with an exocyclic double bond, are really important to inhibit the main protease of coronaviruses [25]. This is in concordance with the data displayed in Table 1 and Table 2, where the sesquiterpenes present higher inhibition values than the benzoxazinoids in general terms. The studies on similar proteins to the RNA replicase and spike protein of coronaviruses is really limited, and there are no small molecules with reported inhibition. However, interesting studies on the M^pro^ of COVID-03 displayed the ability of dibenzyl sulphides (structurally similar to DisOH and DisNH_2_) to link cysteine and histidine [26]. This interaction is observed in the M^pro^ with mimics of the benzoxazinoids tested (DisOH and DisNH_2_) whose main interaction in the binding site involves histide and cysteine. In the last case, Lu et al. remarked the relevance of the sulfur–sulfur interaction [26].

On comparing the standards employed against the SARS-CoV main protease (Figure 2A), it is clear that small differences between the SARS-CoV-2 and SARS-CoV viruses are sufficient to cause differences in ligand binding. According to Xu et al., these two viruses share 96% sequence similarity [27]. The most remarkable example is hydroxychloroquine, which has an inhibition constant in the mM range against SARS-CoV-2 and an inhibition constant of 60 μM against SARS-CoV. Even the site of action of the compound is radically different. The arrangement between the ligand and the target protein is shown in Figure 2B,C, and the different spatial positions in SARS-CoV-2 and SARS-CoV is clear. In the former case, threonine is the main interaction site and this is linked by a hydrogen bond with the terminal hydroxyl group of hydroxychloroquine. In contrast, the SARS-CoV protein binds to the terminal hydroxyl group through a glutamic acid residue and a nitrogen in the structure shows a secondary union with the protein, in this case by a leucine residue. This is a relevant finding according to the experimental results previously published by Liu et al., who reported IC50 values of hydroxychloroquine [28] against SARS-CoV-2 that were ~500 times higher than the IC50 values previously reported against SARS-CoV by Vincent et al. in 2005 [29]. Accordingly, in our computational studies, the IC50 value for hydroxychloroquine against SARS-CoV-2 was only ~200 times higher than for SARS-CoV. On the other hand, azithromycin does not show any activity against the main protease, as one would expect due to the similarities in the previous peptidic inhibitors [30].

In the evaluation of sesquiterpenes and benzoxazinoids, the compounds **Fluor-Reynosin** and **Fluor-Santamarine** are the most promising for the bioassay evaluation. The sites of action for these compounds, i.e., in the main protease and RNA replicase, are the same as for artemisinin. Notwithstanding, the results of an in-depth study on the mode of action of this standard showed that its inhibitory activity is due to a ‘desolvation effect’ caused by a physical impediment toward the protein to be stabilized with solvent in the cytosol. In contrast, **Fluor-Reynosin** and **Fluor-Santamarine**, despite sharing the same action site with artemisinin, are able to establish stronger intermolecular forces. The binding of two histidines instead of one in the case of M^pro^ and one unit of arginine and one valine in the RNA replicase are observed due to the presence of the 4-fluorobenzoate group. Furthermore, this group links with leucine141 and cisteine145, two principal targets in the protease for sesquiterpenoids and benzoxazinoids. Both structures explore new sites of action (Appendix A) that are overlooked by azithromycin, favipiravir and hydroxychloroquine. This situation is exemplified in Appendix A for the main protease.

**Fluor-Cynaro** offers an interesting result in the case of the spike protein due to its long-branched edges, which leads to the establishment of more interactions with the protein than for other ligands. The presence of fluoro-substituents, a high number of carbonyl groups and double-bonded carbons allows more secondary forces to participate in the interaction. The site of action of this compound preferentially enables intermolecular forces with asparagine and glutamic acid, as shown in Appendix A. **Fluor-APO** also presents a remarkable profile along the whole viral protein, and it is more effective than all of the standards in the case of the RNA protease but is best in the case of the spike protein. In addition, it is important to highlight that both compounds can be synthesized in one step in 99% yield by the reaction of the precursor with 4-fluorobenzoyl chloride. Furthermore, both of the precursor natural products (cynaropicrin and **APO**) can be obtained on a multigram scale. [31,32] On considering the results for **DIBOA**, **DIMBOA**, **DDIBOA** and the halogenated derivatives of **DDIBOA**, it is clear that the functionalization of the aromatic ring is not the key aspect. Nevertheless, amide formation, as in the case of **APO**, with the addition of a fluorinated fragment seems to be important. Thus, an extra ring in the structure contributes to a higher binding energy. Furthermore, **Met-4F-Benzo**, the corresponding added fragment, did not show relevant activity when it was not linked to the natural product.

The most promising compounds were considered in the context of Lipinski’s rule in order to evaluate their pharmacological potential in terms of oral bioavailability. The rules and standards are shown in Table 3A. This rule offers a first approach to understand the ADME (absorption, distribution, metabolism and elimination) properties for the compounds selected. The Lipinski ‘rule-of-five’ has had a major impact on the daily practice of medicinal chemistry across the pharmaceutical industry and served as a very useful guideline for orally bioavailable small-molecule drug discovery [33,34]. It can be seen how azithromycin is limited by its high molecular weight and hydrogen-bond acceptors, which could prevent the correct orientation toward protein targets. Nevertheless, all of the sesquiterpenoids and benzoxazinoids shown in Table 1 and Table 2, except for **Fluor-Cynaro**, fulfill the requirements and could be important options in the future development of SARS-CoV-2 inhibitors. This model is dealing only with transport by passive diffusion (a major route for drug molecules permeating thorugh cell membranes). However, diffusion mechanims and complexation with ions also help in the transport. This may present some of the drugs in Table 3B,C as potential drugs with good ADME properties although it is not fulfilling the oral bioavailability Lipinski’s rule. Further analysis may be required in the future to analyze **Fluor-DisNH**, **Fluor-DisOH** and **Fluor-Cynaro** in more details.

### 2.2. Molecular Dynamics Simulations

The MD simulations were run after obtaining the docked positions of the most relevant ligands (**Fluor-Reynosin**, **Fluor-Santamarine** and **Fluor-APO**) (Appendix A). On considering 6LU7, it can be seen from Figure 3 that the RMSD fluctuated by less than 1–1.5 Å for **Fluor-APO** and **Fluor-Reynosin,** and this is consistent with the stable complexes during the whole simulation (50 ns). This finding is also in agreement with the snapshots shown in Appendix A. It is clear from these results that the docked position was fully predicted by molecular docking with these two promising compounds. However, **Fluor-Santamarine** did not give a stable complex in any of the three replicates carried out. According to the score obtained in the docking, both compounds show a similar activity profile, but the different location of the double bond (i.e., exocyclic or endocyclic) seems to determine the stability of the complex at the site of action. The addition of the 4-fluoro benzoate fragment allows to enhance the inhibition of the protein according to the docking score, but isomerism in the double bond allows to generate a long-term complex ligand–protein that shows permanent inhibition. This stability is also observed in the protein–ligand interaction energies in Table 4, which also contains the low Lennard–Jones energy of **Fluor-Santamarine** with the M^pro^ in comparison with the other stable complexes. Furthermore, this stability seems to be directly related to the total and average number of hydrogen bonds per ns. **Fluor-Reynosin** and **Fluor-APO** are able to generate more hydrogen bonds with the protein, thus enhancing the stability of the protein–ligand complex. A structural change in the RMSD is observed during the first 20 ns for **Fluor-Reynosin** and **Fluor-APO,** and these, according to the snapshots shown in Appendix A, are just rotations of the molecule that do not affect the active site. 

In the case of the RNA replicase (6W4B), the RMSD fluctuations for the eudesmanolide derivatives (**Fluor-Reynosin** and **Fluor-Santamarine**) and **Fluor-APO** show stabilization throughout the simulation, with values that do not exceed 1.5 Å (Figure 4 and Appendix A). None of the steps exceed 0.5 Å, and this confirms the stability of the complexes. Nevertheless, **Fluor-APO** experiences a significant continuous variation in geometry throughout the simulation. According to the snapshots (Appendix A), this is not only due to the rotation or rocking of the fluorobenzoate fragment but movement of the whole compound out from the site of action of the protein, thus indicating a kinetically unstable complex. This situation was confirmed by the lower protein–ligand average interaction energy of **Fluor-APO** with the 6W4B protein (RNA replicase) when compared to the other compounds (Table 4). Once again, hydrogen bonding seems to be the main contribution to complex stability. According to the number of hydrogen bonds and their average lifetime, **Fluor-APO** is the worst ligand in terms of inhibiting the action of the RNA replicase in comparison with the other fluorobenzoate derivatives. This finding is consistent with the results shown in Table 1 and Table 2, where it can be seen that the docking energy value for **Fluor-APO** with the RNA replicase is markedly lower than those for the other two compounds analyzed. In addition, **Fluor-APO** also has the lowest average number of hydrogen bonds per nanosecond, which is consistent with the continuous increase in the RMSD value as the ligand moves away.

The spike (6M0J) receptor-binding protein–ligand complexes were also analyzed. It can be seen from Appendix A that the **Fluor-Santamarine** and **Fluor-Reynosin** complexes are stable after 50 ns, while **Fluor-APO** is relatively unstable with an RMSD fluctuation above 9 Å, which means that the ligand position is not stable at that docking point. The score values from the docking studies show that **Fluor-APO** is a promising compound, but the MD simulations show a kinetically unstable complex. This is graphically represented in the snapshots, where **Fluor-APO** changes its position markedly with respect to the spike protein (Appendix A). This situation is consistent with the energy values of the ligands, where the protein–ligand energy differs between **Fluor-APO** and the other two ligands by a factor of greater than fifteen. The number of hydrogen bonds is a relevant parameter in terms of the energy and stability of the complex and, in this case, complexes with the 6M0J protein seem to generate structures with lower stability in comparison to other proteins (Table 4. The number of hydrogen bonds is reduced dramatically, with **Fluor-Reynosin** showing only 1674 H-bonds. This value is extremely small in comparison with the numbers of hydrogen bonds generated in the cases of the other proteins, although the protein–ligand interaction energies have comparable values. It appears that other intermolecular forces that have more profound energetic implications must be involved in the interaction with the protein to contribute to the stability of the **Fluor-Reynosin** and **Fluor-Santamarine** complexes.

## 3. Conclusions

In silico studies such as molecular docking and dynamic methods represent a relevant and rapid advance in the search for new drugs from derivatives of natural compounds against SARS-CoV-2. The results reported here highlight the potential use of sesquiterpenoids and benzoxazinoids to fight this virus. The molecules evaluated in this study have a different site of action when compared with compounds from the same families that have previously shown activity against the virus in preliminary studies. Furthermore, the results of the molecular dynamics studies corroborated the docking results, thus showing the stability of the protein–ligand complex by the RMSD fluctuations—especially the complexes with the M^pro^ and RNA replicase. Our team is currently analyzing and selecting possible candidates based on the docking scores and physicochemical properties in an effort to identify the best candidates for molecular dynamics studies. The results reported here indicate that the addition of the 4-fluorobenzoate fragment to the natural products enhances their potential against all of the proteins tested. This option would allow the production of a large number of drug leads, and it would be possible to synthesize the most remarkable compounds (**Fluor-Reynosin**, **Fluor-Santamarine** and **Fluor-APO**) in just one step.

## 4. Materials and Methods

### 4.1. Molecular Docking Studies

The 2D structures of the assayed compounds were generated with ChemBioDraw 20.0 and were converted to 3D structures with GaussView 6.0.16 software (Wallingford, CT, USA). Proteins were obtained from the Protein Data Bank (www.rcsb.org, accessed on 1 April 2020). The proteins selected were 2GTB (main protease of SARS-CoV), 6LU7 (main protease of SARS-CoV-2), 6W4B (RNA replicase of SARS-CoV-2) and 6M0J (spike receptor binding of SARS-CoV-2). A grid box (120 × 120 × 120 Å) was generated and centered on the proteins. Kollman charges were applied to each protein to simulate the electrostatic potential of amino acids. AutoDockTools (v. 1.5.6) was employed to define the previous steps. DFT B3LYP/6-311G(d,p) minimization was employed prior to carrying out the docking. Autodock 4.2 and the Lamarckian GA algorithm with 20 GA runs were employed to develop the local docking, with a value of 1.0 used as the variance of the Cauchy distribution for gene mutations. All calculations correspond to the most populated cluster, with at least three members that fulfill an RMSD tolerance below 2.000 Å (Appendix A). Discovery Studio Visualizer 19.0 was used for the refinement of the docking results. Chemical Identifier Resolver [35] was employed for the calculation of properties related to Lipinski’s rule.

### 4.2. Molecular Dynamics Simulation

The studies were carried out starting from the minimum energy protein–ligand conformation obtained from the previous molecular docking studies. GROMACS (2019.6 version) was employed in conjunction with CHARMM36 force-field (march-2019) and SPCE water model. The ligand topologies and parameters were obtained using the SwissParam server (www.swissparam.ch, accessed on 5 June 2021) [36]. A dodecahedral box was generated and the protein–ligand complexes (6LU7, 6W4 and 6M0J) were at least 1 nm from the edges of the box, with a distance of at least 2 nm between periodic images of the protein in order to fulfill the minimum image convention. A 0.1 M NaCl concentration was simulated in the system to mimic physiological conditions. An energy minimization was applied until the maximum force was less than 10 kJ/mol. The system was then equilibrated for 0.1 ns with 2 fs per step at 300 K using canonical equilibration. Equilibration of the pressure was then carried out by the isothermal–isobaric method using the Parrinello–Rahman barostat. The system was equilibrated for 0.1 ns, also with 2 fs per step, at 300 K. The full equilibrated system was submitted to a molecular dynamics simulation for 50 ns with 2 fs per step. Correction of the trajectory was carried out by protein recentering within the dodecahedral box. Snapshots of the trajectory were collected every 10 ns. The average number of hydrogen bonds and average distance of these bonds were calculated using a 0.35 nm cut-off distance.

## Figures and Tables

**Figure 1 toxins-14-00599-f001:**
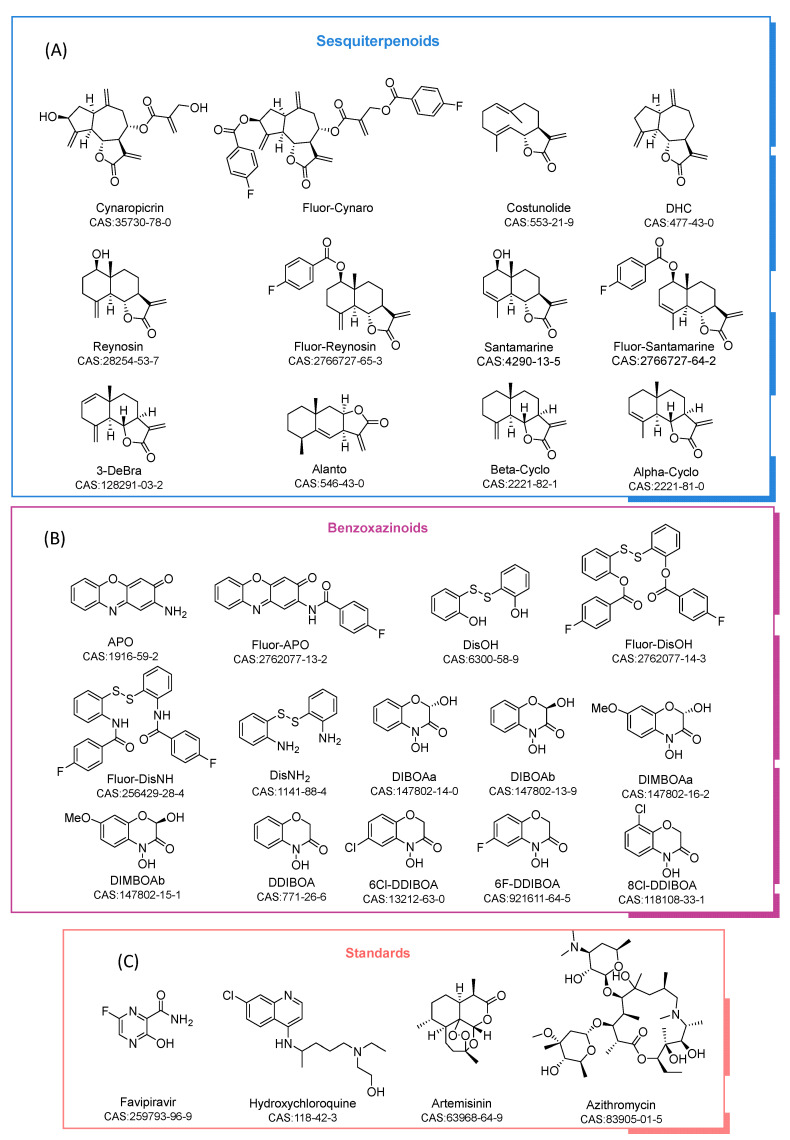
(**A**) Sesquiterpenoids tested in the molecular docking analysis. (**B**) Benzoxazinoids tested and (**C)** standards employed.

**Figure 2 toxins-14-00599-f002:**
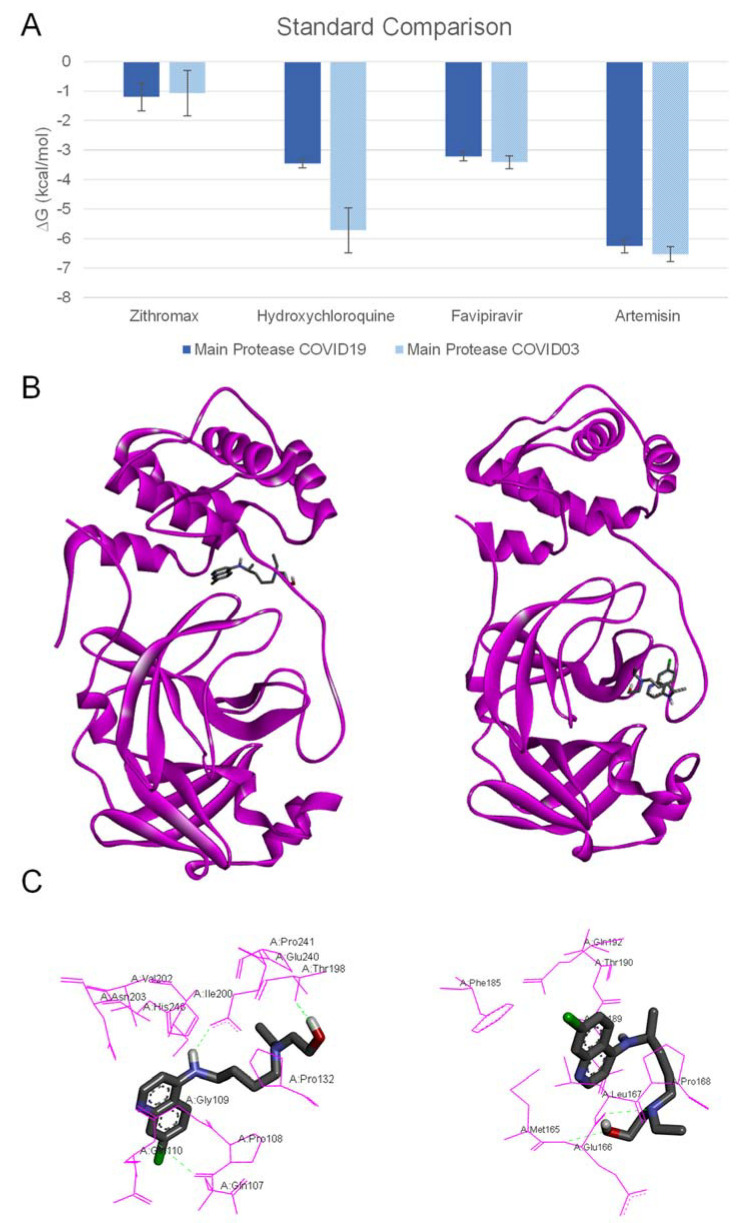
(**A**) Comparison between binding energy values of standard compounds against M^pro^ of SARS-CoV and M^pro^ SARS-CoV-2. (**B Left**) Site of action of hydroxychloroquine on M^pro^ SARS-CoV-2. (**B Right**) Site of action of hydroxychloroquine on M^pro^ SARS-CoV. (**C Left**) Amino acid residues that establish intermolecular forces with hydroxychloroquine on M^pro^ SARS-CoV-2. (**C Right**) Amino acid residues that establish intermolecular forces with hydroxychloroquine on M^pro^ SARS-CoV.

**Figure 3 toxins-14-00599-f003:**
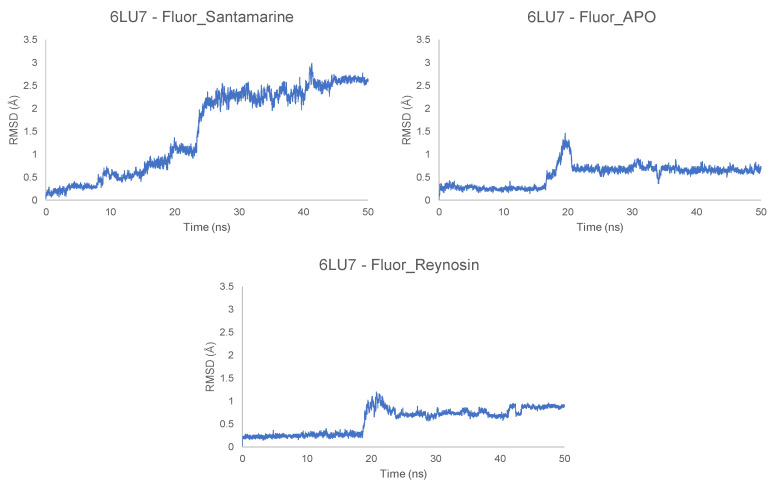
Root mean square deviation (RMSD) of the different ligands in the protein–ligand complex with the main protease (6LU7) of SARS-CoV-2.

**Figure 4 toxins-14-00599-f004:**
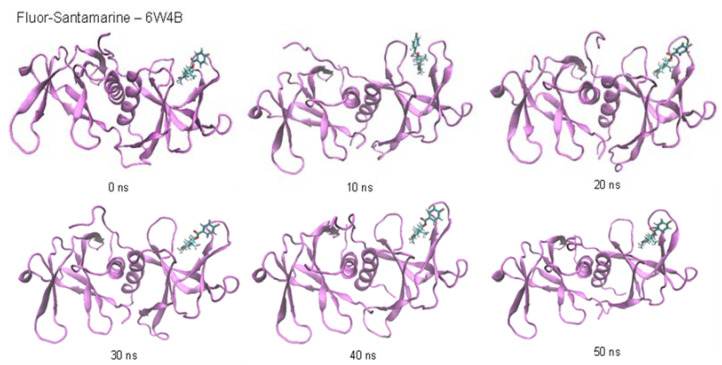
Root mean square deviation (RMSD) of Fluor-Santamatine in the protein–ligand complex with the RNA replicase (6W4B) of SARS-CoV-2.

**Table 1 toxins-14-00599-t001:** Binding energy values of sesquiterpenoids selected in the study on Mpro, RNA replicase and spike protein of SARS-CoV-2.

Compounds	ΔG (Kcal/mol)		
	Main Protease	RNA Replicase	Spike Protein
Azithromycin	−1.20 ± 0.47	−0.76 ± 0.88	−4.64 ± 0.78
Hydroxychloroquine	−3.45 ± 0.16	−2.67 ± 0.81	−4.29 ± 0.76
Favipiravir	−3.21 ± 0.16	−3.58 ± 0.24	−3.93 ± 0.42
Artemisinin	−6.25 ± 0.23	−6.07 ± 0.07	−5.96 ± 0.19
Cynaropicrin	−3.49 ± 0.07	−4.02 ± 0.25	−4.19 ± 0.28
Met-4F-Benzo	−5.32 ± 0.64	−5.71 ± 0.93	−5.81 ± 0.48
Fluor-Cynaro	−3.97 ± 1.01	−5.56 ± 0.73	−8.13 ± 1.08
Costunolide	−6.11 ± 0.41	−5.77 ± 0.20	−6.15 ± 0.20
DHC	−6.08 ± 0.30	−5.57 ± 0.34	−6.00 ± 0.28
Reynosin	−5.54 ± 0.48	−5.92 ± 0.41	−6.11 ± 0.18
Santamarine	−5.81 ± 0.59	−5.97 ± 0.48	−6.12 ± 0.31
Fluor-Reynosin	−7.37 ± 0.50	−7.10 ± 0.93	−7.89 ± 0.77
Fluor-Santamarine	−7.77 ± 0.77	−6.35 ± 0.54	−7.68 ± 0.74
Alanto	−5.82 ± 0.22	−6.57 ± 0.26	−6.46 ± 0.17
Alpha-Cyclo	−5.99 ± 0.37	−5.95 ± 0.36	−6.20 ± 0.25
Beta-Cyclo	−5.98 ± 0.49	−6.02 ± 0.52	−6.20 ± 0.29
3-DeBra	−6.36 ± 0.37	−6.04 ± 0.37	−6.19 ± 0.25

**Table 2 toxins-14-00599-t002:** Binding energy values of benzoxazinoids selected in the study on M^pro^, RNA replicase and the spike protein of SARS-CoV-2.

Compounds	ΔG (Kcal/mol)		
	Main Protease	RNA Replicase	Spike Protein
Azithromycin	−1.20 ± 0.47	−0.76 ± 0.88	−4.64 ± 0.78
Hydroxychloroquine	−3.45 ± 0.16	−2.67 ± 0.81	−4.29 ± 0.76
Favipiravir	−3.21 ± 0.16	−3.58 ± 0.24	−3.93 ± 0.42
Artemisinin	−6.25 ± 0.23	−6.07 ± 0.07	−5.96 ± 0.19
Met-4F-Benzo	−3.49 ± 0.08	−4.02 ± 0.25	−4.19 ± 0.28
APO	−5.13 ± 0.31	−5.93 ± 0.66	−5.52 ± 0.24
DisOH	−4.84 ± 0.69	−4.74 ± 0.84	−4.66 ± 0.59
DisNH_2_	−4.48 ± 0.22	−4.68 ± 0.44	−4.88 ± 0.38
Fluor-APO	−6.01 ± 0.53	−6.08 ± 0.41	−7.79 ± 0.88
Fluor-DisOH	−5.71 ± 1.36	−4.67 ± 0.69	−5.01 ± 0.86
Fluor-DisNH	−4.45 ± 1.31	−5.77 ± 0.55	−5.91 ± 0.93
DIBOAa	−4.05 ± 0.28	−4.05 ± 0.34	−4.94 ± 0.33
DIBOAb	−3.90 ± 0.17	−4.50 ± 0.49	−4.33 ± 0.28
DIMBOAa	−3.93 ± 0.17	−4.03 ± 0.38	−4.61 ± 0.30
DIMBOAb	−3.91 ± 0.13	−3.71 ± 0.47	−4.53 ± 0.36
DDIBOA	−4.12 ± 0.17	−4.19 ± 0.19	−4.41 ± 0.27
6Cl-DDIBOA	−4.42 ± 0.11	−4.28 ± 0.24	−4.77 ± 0.21
6F-DDIBOA	−4.07 ± 0.28	−4.30 ± 0.36	−4.28 ± 0.19
6F-DDIBOA	−4.46 ± 0.22	−4.57 ± 0.38	−4.57 ± 0.18

**Table 3 toxins-14-00599-t003:** Lipinski’s rules evaluation of compounds evaluated in molecular docking.

**(A).** Lipinski’s Rules for standard compounds tested.				
**No**	**Standard**	**Molecular Formula**	**Lipinski’s rule of 5**	
			**Properties**	**Value**
**1**	6-fluoropyrazine-2-carboxamide (Favipiravir)	C_5_H_4_FN_3_O	M.W. (≤500 amu)	141.11
			cLog *P* (≤5)	–0.50873
			H-bond donors (≤5)	1
			H-bond acceptors (≤10)	5
			Violations	0
**2**	Hydroxychloroquine	C_18_H_26_ClN_3_O	M.W. (≤500 amu)	335.88
			cLog *P* (≤5)	4.11588
			H-bond donors (≤5)	2
			H-bond acceptors (≤10)	4
			Violations	0
**3**	Artemisinin	C_15_H_22_O_5_	M.W. (≤500 amu)	282.34
			cLog *P* (≤5)	2.71630
			H-bond donors (≤5)	0
			H-bond acceptors (≤10)	5
			Violations	0
**4**	Azithromycin	C_38_H_72_N_2_O_12_	M.W. (≤500 amu)	749.00
			cLog *P* (≤5)	2.63825
			H-bond donors (≤5)	5
			H-bond acceptors (≤10)	14
			Violations	2
**(B).** Lipinski’s Rules for the most relevant sesquiterpenoid compounds tested.				
**No**	**Sesquiterpenoids**	**Molecular Formula**	**Lipinski’s rule of 5**	
			**Properties**	**Value**
**1**	Cynaropicrin	C_19_H_22_O_6_	M.W. (≤500 amu)	346.38
			cLog *P* (≤5)	0.045825
			H-bond donors (≤5)	2
			H-bond acceptors (≤10)	6
			Violations	0
**2**	3,3’-di(4’-fluorobenzoyloxy)cynaropicrin (Fluor-Cynaro)	C_33_H_28_F_2_O_8_	M.W. (≤500 amu)	590.58
			cLog *P* (≤5)	5.82662
			H-bond donors (≤5)	0
			H-bond acceptors (≤10)	10
			Violations	2
**3**	Costunolide	C_15_H_20_O_2_	M.W. (≤500 amu)	232.32
			cLog *P* (≤5)	3.79
			H-bond donors (≤5)	0
			H-bond acceptors (≤10)	2
			Violations	0
**4**	Dehydrocostuslactone (DHC)	C_15_H_18_O_2_	M.W. (≤500 amu)	230.31
			cLog *P* (≤5)	2.786
			H-bond donors (≤5)	0
			H-bond acceptors (≤10)	2
			Violations	0
**5**	Reynosin	C_15_H_20_O_3_	M.W. (≤500 amu)	248.32
			cLog *P* (≤5)	1.183
			H-bond donors (≤5)	1
			H-bond acceptors (≤10)	3
			Violations	0
**6**	1-(4’-fluorobenzoyloxy)reynosin (Fluor-Reynosin)	C_22_H_23_FO_4_	M.W. (≤500 amu)	370.42
			cLog *P* (≤5)	4.201
			H-bond donors (≤5)	0
			H-bond acceptors (≤10)	5
			Violations	0
**7**	Santamarine	C_15_H_20_O_3_	M.W. (≤500 amu)	248.32
			cLog *P* (≤5)	1.183
			H-bond donors (≤5)	1
			H-bond acceptors (≤10)	3
			Violations	0
**8**	1-(4-fluorobenzoyloxy)santamarine (Fluor-Santamarine)	C_22_H_23_FO_4_	M.W. (≤500 amu)	370.42
			cLog *P* (≤5)	4.201
			H-bond donors (≤5)	0
			H-bond acceptors (≤10)	5
			Violations	0
**9**	Alantolactone (Alanto)	C_15_H_20_O_2_	M.W. (≤500 amu)	232.32
			cLog *P* (≤5)	3.27
			H-bond donors (≤5)	0
			H-bond acceptors (≤10)	2
			Violations	0
**10**	β-cyclocostunolide (Beta-Cyclo)	C_15_H_20_O_2_	M.W. (≤500 amu)	232.32
			cLog *P* (≤5)	3.27
			H-bond donors (≤5)	0
			H-bond acceptors (≤10)	2
			Violations	0
**11**	α-cyclocostunolide (Alpha-Cyclo)	C_15_H_20_O_2_	M.W. (≤500 amu)	232.32
			cLog *P* (≤5)	3.27
			H-bond donors (≤5)	0
			H-bond acceptors (≤10)	2
			Violations	0
**12**	3-deoxybrachylaenolide (3-DeBra)	C_15_H_16_O_3_	M.W. (≤500 amu)	244.29
			cLog *P* (≤5)	1.024
			H-bond donors (≤5)	0
			H-bond acceptors (≤10)	3
			Violations	0
**(C).** Lipinski’s Rules for the most relevant aminophenoxazinoids tested.				
**No**	**Benzoxazinoids**	**Molecular Formula**	**Lipinski’s rule of 5**	
			**Properties**	**Value**
**13**	2-amino-3*H*-phenoxazin-3-one (APO)	C_12_H_8_N_2_O_2_	M.W. (≤500 amu)	212.21
			cLog *P* (≤5)	1.13575
			H-bond donors (≤5)	1
			H-bond acceptors (≤10)	4
			Violations	0
**14**	4-fluoro-*N*-(3-oxo-3*H*-phenoxazin-2-yl)benzamide (Fluor-APO)	C_19_H_11_FN_2_O_3_	M.W. (≤500 amu)	334.31
			cLog *P* (≤5)	2.97045
			H-bond donors (≤5)	1
			H-bond acceptors (≤10)	6
			Violations	0
**15**	2,2′-disulfanediyldiphenol (DisOH)	C_12_H_10_O_2_S_2_	M.W. (≤500 amu)	250.33
			cLog *P* (≤5)	3.0194
			H-bond donors (≤5)	2
			H-bond acceptors (≤10)	2
			Violations	0
**16**	disulfanediylbis(2,1-phenylene) bis(4-fluorobenzoate) (Fluor-DisOH)	C_26_H_16_F_2_O_4_S_2_	M.W. (≤500 amu)	494.53
			cLog *P* (≤5)	7.2229
			H-bond donors (≤5)	0
			H-bond acceptors (≤10)	6
			Violations	1
**17**	2,2′-dithiodianiline (DisNH_2_)	C_12_H_12_N_2_S_2_	M.W. (≤500 amu)	248.36
			cLog *P* (≤5)	2.736
			H-bond donors (≤5)	2
			H-bond acceptors (≤10)	2
			Violations	0
**18**	*N*,*N′*-(disulfanediylbis(2,1-phenylene))bis(4-fluorobenzamide) (Fluor-DisNH)	C_26_H_18_F_2_N_2_O_2_S_2_	M.W. (≤500 amu)	492.56
			cLog *P* (≤5)	5.06192
			H-bond donors (≤5)	2
			H-bond acceptors (≤10)	4
			Violations	1

**Table 4 toxins-14-00599-t004:** Relevant energy values and intermolecular interactions of every protein with a ligand, surrounded by ions and water molecules.

**Protein–LIG Energy (kJ/mol) Lennard–Jones**								
**6M0J**			**6LU7**			**6W4B**		
Fluor-APO	Fluor-Santamarine	Fluor-Reynosin	Fluor-APO	Fluor-Santamarine	Fluor-Reynosin	Fluor-APO	Fluor-Santamarine	Fluor-Reynosin
−8.846 ± 4.279	−123.866 ± 3.611	−113.915 ± 2.960	−105.696 ± 2.446	−68.819 ± 3.583	−126.618 ± 2.007	−54.026 ± 2.800	−118.928 ± 1.982	−125.207 ± 10.865
**Protein–LIG Total Number of H-Bonds along 50 ns**								
**6M0J**			**6LU7**			**6W4B**		
Fluor-APO	Fluor-Santamarine	Fluor-Reynosin	Fluor-APO	Fluor-Santamarine	Fluor-Reynosin	Fluor-APO	Fluor-Santamarine	Fluor-Reynosin
355	1267	1674	3600	2490	3708	1293	3571	2385
**Protein–LIG Average Number of H-Bonds per ns**								
**6M0J**			**6LU7**			**6W4B**		
Fluor-APO	Fluor-Santamarine	Fluor-Reynosin	Fluor-APO	Fluor-Santamarine	Fluor-Reynosin	Fluor-APO	Fluor-Santamarine	Fluor-Reynosin
0.07	0.25	0.33	0.72	0.50	0.74	0.26	0.91	0.48
**Protein–LIG Average Distance of H-Bonds (nm)**								
**6M0J**			**6LU7**			**6W4B**		
Fluor-APO	Fluor-Santamarine	Fluor-Reynosin	Fluor-APO	Fluor-Santamarine	Fluor-Reynosin	Fluor-APO	Fluor-Santamarine	Fluor-Reynosin
0.2925	0.3075	0.2825	0.2875	0.2975	0.3125	0.3275	0.2875	0.2825
**Protein–LIG Lifetime of H-Bonds (ps)**								
**6M0J**			**6LU7**			**6W4B**		
Fluor-APO	Fluor-Santamarine	Fluor-Reynosin	Fluor-APO	Fluor-Santamarine	Fluor-Reynosin	Fluor-APO	Fluor-Santamarine	Fluor-Reynosin
19.63	18.08	14.03	65.95	24.34	19.81	27.57	74.94	75.55

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
