# Peer review of "In Silico Evaluation of Sesquiterpenes and Benzoxazinoids Phytotoxins against Mpro, RNA Replicase and Spike Protein of SARS-CoV-2 by Molecular Dynamics. Inspired by Nature"

_toxins, 2022, doi:10.3390/toxins14090599_

Round 1

Reviewer 1 Report

The author presented the data with no in vitro or in vivo evidence. The abstract does not reflect the results or discussion, and it must be rewritten with detailed information from the results, discussion and conclusion. The discussion in the manuscript is completely missing. In the result section, Figures 2 and 3 can be well-presented in Table format. The authors presented only results (I would say), and the number of references is poor. Without proper discussion and additional analysis of similar studies, the paper is not acceptable.  The Title is also incorrect. 

Author Response

Dear reviewer,

Thank you very much for your comments. We have modified every aspect according to the changes that you proposed:

The abstract does not reflect the results or discussion, and it must be rewritten with detailed information from the results, discussion and conclusion.

Abstract have been re-written in order to include results, discussion and conclusions related with the experiment that have been carried out.

The discussion in the manuscript is completely missing.

Several paragraphs and sentences have been included all over the manuscript in order to enhance the discussion of the results. For example Line 135-146 include a comparison of the in silico results obtained with similar molecules tested on the proteins. Line 212-229 show a discussion about the oral bioavailability possibility according to Lipinski’s rule. In addition, comments in the MD section have been also included for more discussion.

In the result section, Figures 2 and 3 can be well-presented in Table format.

Done. Figure 2 and 3 are now tables.

The authors presented only results (I would say), and the number of references is poor.

Number of references have been increased together with discussion of the data.

The Title is also incorrect.

Authors think that title is covering main objectives of the work (evaluation of specific compounds against three proteins of SARS-CoV-2), techniques employed in the work (in silico and Molecular Dynamics), and the main source of the compounds tested (sesquiterpenes and benzoxacinoids, inspired by nature).

Reviewer 2 Report

The manuscript “In silico evaluation of sesquiterpenes and benzoxazinoids phytotoxins against Mpro, RNA replicase and spike protein of SARS-CoV-2 by molecular dynamics. Inspired by nature.” [toxins-1878158] submitted in a double blind peer review describes a molecular docking of a couple of natural and artificial sesquiterpenoids as well as benzoxazinoids protease, RNA replicase and spike glycoprotein of SARS-CoV-2. A large number of calculations were carried out for this purpose and the data obtained were statistically evaluated. The calculations have been carried out using state-of-the-art methods. The results appear to make sense. However, there are no experimental results at the molecular level, which are / can be used for comparison.

The last point surprised the reviewer somewhat, since the expertise of the reviewer is also in this area. It is therefore difficult for the reviewer to write a detailed review of the details of the computer-aided calculations. The report therefore relates more to the selected natural substances and derivatives and to possible applications.

The presented overall work seems, however, quite well planned and performed. The reported results sound perspicuous and are quite well described. The results possess some importance in furthering our knowledge of binding behavior between possible drug candidates for treatment of SARS CoV-2 and some relevant proteins in SARS CoV-2. However, there are a few minor weaknesses in the presentation that should be revised by the authors (see comments).

The manuscript is hence of great interest in the fields of Theoretical Chemistry and Natural Product Chemistry, as well as in Pharmaceutical Chemistry and
Virology. However, the reviewer has a few (minor) comments that should be considered by the authors before the manuscript can be accepted for publication in "Toxines".

(Minor) Comments:

1) The selection of the 12 sesquiterpene lactones and 14 benzoxazinoids is explained in detail. However, the naming of the compounds is carried out with many abbreviations, which are only partially explained. It would therefore make sense to state the CAS numbers of the compounds at least once (e.g. in Figure 1).

2) The presented binding studies are based on data generated in silico. It would be interesting to have in the "discussion" comparisons to measured data from previous binding studies performed by the same sesquiterpene lactones and benzoxazinoids on similar proteins. This could be of particular interest for prospectively effective candidates.

Author Response

Dear reviewer,

Thank you very much for your nice comments. I’m glad that you enjoyed our work. We have modified every aspect according to the minor changes that you proposed:

The selection of the 12 sesquiterpene lactones and 14 benzoxazinoids is explained in detail. However, the naming of the compounds is carried out with many abbreviations, which are only partially explained. It would therefore make sense to state the CAS numbers of the compounds at least once (e.g. in Figure 1)

Figure 1 have been modified and now you can find the CAS number of every compound teste in silico

The presented binding studies are based on data generated in silico. It would be interesting to have in the "discussion" comparisons to measured data from previous binding studies performed by the same sesquiterpene lactones and benzoxazinoids on similar proteins. This could be of particular interest for prospectively effective candidates

It has been added a paragraph (Line 134-146) discussing similarities about interactions and residues between similar molecules and SARS-CoV proteins. Sulfur containing compounds seems to be really active due to his link to histidines and cysteines. On the other Michael additions, related with a,b unsaturated lactone fragment, have demonstrated his efficacy to inhibit these proteins.    

Reviewer 3 Report

The manuscript submitted to Toxins, can be accepted to publication after some minor changes and improvements, as required bellow:

- primary references for the PDB entrees used in the docking protocols, must be added;

- reference for Lipinski rule of five, should be added and discussed rather in term of oral bioavailability;

- reference for the computed method for logP computations and software used, must be mentioned;

- link between docking score and IC50 should be explained.

Author Response

Dear reviewer,

Thank you very much for your nice comments. I’m glad that you enjoyed our work. We have modified every aspect according to the minor changes that you proposed:

Primary references for the PDB entrees used in the docking protocols, must be added

References for PDB entrees and publications related with the discover of the proteins have been added. They can be found in the reference list, [8-13].

Reference for Lipinski rule of five, should be added and discussed rather in term of oral bioavailability.

References have been added [33,34]. Discussion about oral bioavailability has been extended. Lines 211-228.

Reference for the computed method for logP computations and software used, must be mentioned.

A reference and description about the software have been added in the experimental part. Chemical Identifier Resolver [35] was employed for the calculation of properties related to Lipinski’s rule. This software is online available and it has been created by USA National Cancer Institute.

Link between docking score and IC50 should be explained.

There is a direct connection between docking score (ΔG calculated by the software) and the IC50 by the following equations:

ΔG=R·T·Ln(Kd/c0)

Kd=1/Ka 

IC50=([Prot]/2)+Ka

Where, c0 is the standard reference concentration (usually 1 mol/L), Kd is the dissociation constant, Ka is the association constant and [Prot] is the total concentration of the protein (usually 1 mol/L) for simulations. This is the approach carried out by Autodock software. However, this just an stimation due to lack of other interactions between the guest and the protein. For example, no counterions and water molecules are taken into account. Furthermore, it is a simulation of one protein with one guest in an empty space, withouth other molecules in the media.

Reviewer 4 Report

In my opinion, this study presents up-to-date information that can be used in the treatment of covid 19. The strengths are represented by the computational analyzes and the use of some natural molecules in this study. The figures are clear. In general, the text is clear and concise, but in some places the author could use a more accessible language for a wider and more transdisciplinary audience. In the supplementary material, the title of the tables must be placed above the tables and the abbreviated terms used in the tables must be explained below the table. I have only one specific recommendation, to specify in the first part of the L50 - 65 results if the studied molecules also have antiviral properties according to the specialized literature. Also, please pay attention to the punctuation  ``cytotoxicity.[5–7]'' L54, please check L61, L65, etc. throughout the text.

Author Response

Dear reviewer,

Thank you very much for your nice comments. I’m glad that you enjoyed our work. We have modified every aspect according to the minor changes that you proposed:

In the supplementary material, the title of the tables must be placed above the tables and the abbreviated terms used in the tables must be explained below the table.

Fixed. Now, titles are above the tables.

I have only one specific recommendation, to specify in the first part of the L50 - 65 results if the studied molecules also have antiviral properties according to the specialized literature. Also, please pay attention to the punctuation ``cytotoxicity.[5–7]'' L54, please check L61, L65, etc. throughout the text.

It has been added a sentence about antiviral activities of the compounds tested. Lines 80-84. Punctuation have been checked all over the manuscript. Thank you very much for the advice.

Round 2

Reviewer 1 Report

Thanks to the author(s) for the revision. The manuscript is improved, but the formatting of the manuscript is lost.

The abstract now contains errors like: "Compounds tested in vitro reach as candidates from natural sources to fight this virus - especially given the ease of access to fluoro-substituted derivatives that can be obtained on a gram scale"

As I checked, the author(s) did not conduct any in vitro or in vivo studies. So, the sentence is fully incorrect, and it must be rewritten.

Moreover, the abstract does not contain any conclusion or overview of their outcome. It should be incorporated. 

Author Response

Dear reviewer,

I'm glad that you are satisfied with the new comments that we added. Furthermore, we would like to thanks your new comments that will improve the final manuscript.

- We are sorry about the format. It has been improved now. Specifically, Figures and Tables that were moved in the conversion from Word to PDF.

- In vitro has been changed to In silico. Yo are absolutely right, it was a mistake.

- Abstract has been modified once again to include more aspects about conclusions. It is highlighted how the molecules interact with the proteins for the inhibition and the relevance of 4-fluorobenzoate fragment in the derivatization of natural products to enhance activity and complex stability.

Thank you very much for your time to review the manuscript.

Best regards.